# On Completeness-aware Concept-Based Explanations in Deep Neural Networks

Chih-Kuan Yeh[1], Been Kim[2], Sercan Ö. Arık[3], Chun-Liang Li[3],
Tomas Pfister[3], and Pradeep Ravikumar[1]

[1]Machine Learning Department, Carnegie Mellon University
[2]Google Brain
[3]Google Cloud AI

## Abstract

Human explanations of high-level decisions are often expressed in terms of key concepts the decisions are based on. In this paper, we study such concept-based explainability for Deep Neural Networks (DNNs). First, we define the notion of *completeness*, which quantifies how sufficient a particular set of concepts is in explaining a model's prediction behavior based on the assumption that complete concept scores are sufficient statistics of the model prediction. Next, we propose a concept discovery method that aims to infer a complete set of concepts that are additionally encouraged to be interpretable, which addresses the limitations of existing methods on concept explanations. To define an importance score for each discovered concept, we adapt game-theoretic notions to aggregate over sets and propose *ConceptSHAP*. Via proposed metrics and user studies, on a synthetic dataset with apriori-known concept explanations, as well as on real-world image and language datasets, we validate the effectiveness of our method in finding concepts that are both complete in explaining the decisions and interpretable.[1]

## 1 Introduction

The lack of explainability of deep neural networks (DNNs) arguably hampers their full potential for real-world impact. Explanations can help domain experts better understand rationales behind the model decisions, identify systematic failure cases, and potentially provide feedback to model builders for improvements. Most commonly-used methods for DNNs explain each prediction by quantifying the importance of each input feature [Ribeiro et al., 2016, Lundberg and Lee, 2017]. One caveat with such explanations is that they typically focus on the local behavior for each data point, rather than globally explaining how the model reasons. Besides, the weighted input features are not necessarily the most intuitive explanations for human understanding, particularly when using low-level features such as raw pixel values. In contrast, human reasoning often comprise "concept-based thinking," extracting similarities from numerous examples and grouping them systematically based on their resemblance [Armstrong et al., 1983, Tenenbaum, 1999]. It is thus of interest to develop such "concept-based explanations" to characterize the global behavior of a DNN in a way understandable to humans, explaining how DNNs use concepts in arriving at particular decisions.

A few recent studies have focused on bringing such concept-based explainability to DNNs, largely based on the common implicit assumption that the concepts lie in low-dimensional subspaces of some intermediate DNN activations. Via supervised training based on labeled concepts, TCAV [Kim et al., 2018] trains linear concept classifiers to derive concept vectors, and uses how sensitive predictions are to these vectors (via directional derivatives) to measure the importance of a concept with respect to a specific class. Zhou et al. [2018] considers the decomposition of model predictions in terms of projections onto concept vectors. Instead of human-labeled concept data, Ghorbani et al.

[2019] employs k-means clustering of super-pixel segmentations of images to discover concepts. Bouchacourt and Denoyer [2019] proposes a Bayesian generative model involving concept vectors. One drawback of these approaches is that they do not take into account *how much* each concept plays a role in the prediction. In particular, selecting a set of concepts salient to a particular class does not guarantee that these concepts are sufficient in explaining the prediction. The notion of sufficiency is also referred to as "completeness" of explanations, as in [Gilpin et al., 2018, Yang et al., 2019]. This motivates the following key questions: Is there an unsupervised approach to extract concepts that are sufficiently predictive of a DNN's decisions? If so, how can we measure this sufficiency?

In this paper, we propose such a completeness score for concept-based explanations. Our metric can be applied to a set of concept vectors that lie in a subspace of some intermediate DNN activations, which is a general assumption in previous work in this context [Kim et al., 2018, Zhou et al., 2018]. Intuitively speaking, a set of "complete" concepts can fully explain the prediction of the underlying model. By further assuming that for a complete set of concepts, the projections of activations onto the concepts are a sufficient statistic for the prediction of the model, we may measure the "completion" of the concepts by the accuracy of the model just given these concept based sufficient statistics. For concept discovery, we propose a novel algorithm, which could also be viewed as optimizing a surrogate likelihood of the concept-based data generation process, motivated by topic modeling [Blei et al., 2003]. To ensure that the discovered complete concepts are also coherent (distinct from other concepts) and semantically meaningful, we further introduce an interpretability regularizer.

Beyond concept discovery, we also propose a score, *ConceptSHAP*, for quantification of concept attributions as contextualized importance. ConceptSHAP uniquely satisfies a key set of axioms involving the contribution of each concept to the completeness score [Shapley, 1988, Lundberg and Lee, 2017]. We also propose a class-specific version of ConceptSHAP that decomposes it with respect to each class in multi-class classification. This can be used to find class-specific concepts that contribute the most to a specific class. To verify the effectiveness of our *automated* completeness-aware concept discovery method, we create a synthetic dataset with apriori-known ground truth concepts. We show that our approach outperforms all compared methods in correct retrieval of the concepts as well as in terms of its coherency via a user study. Lastly, we demonstrate how our concept discovery algorithm provides additional insights into the behavior of DNN models on both image and language real-world datasets.

## 2    Related Work

Most post-hoc interpretability methods fall under the categories: (a) feature-based explanation methods, that attribute the decision to important input features [Ribeiro et al., 2016, Lundberg and Lee, 2017, Smilkov et al., 2017, Chen et al., 2018], (b) sample-based explanation methods, that attribute the decision to previously observed samples [Koh and Liang, 2017, Yeh et al., 2018, Khanna et al., 2019, Arik and Pfister, 2019], and (c) counterfactual-based explanation methods, which answers the question: "what to alter in the current input to change the outcome of the model" [Wachter et al., 2017, Dhurandhar et al., 2018, Hendricks et al., 2018, van der Waa et al., 2018, Goyal et al., 2019a, Joshi et al., 2019, Poyiadzi et al., 2020].Recent work has also focused on *evaluations* of explanations, ranging from human-centric evaluations [Lundberg and Lee, 2017, Kim et al., 2018, Lage et al., 2019] to functionally-grounded evaluations [Samek et al., 2016, Kim et al., 2016, Ancona et al., 2017, Yeh et al., 2019, Yang et al., 2019, Guidotti et al., 2018, Hooker et al., 2018, Yang and Kim, 2019]. Our work provides an evaluation of concept explanations based on the completeness criteria, which is related to the 'fidelity' [Guidotti et al., 2018].

Our work is related to methods that learn semantically-meaningful latent variables. Some use dimensionality reduction methods [Chan et al., 2015, Kingma and Welling, 2013], while others uncover higher level human-relatable concepts by dimensionality reduction (e.g. for speech [Chorowski et al., 2019] and language [Radford et al., 2017, Grover et al., 2019]). More recently Locatello et al. [2018] shows that meaningful latent dimensions cannot be acquired in a completely unsupervised setting, implying the necessity of inductive biases for discovering meaningful latent dimensions. Our work uses indirect supervision from the classifier of interest to discover semantically meaningful latent dimensions. Chen et al. [2019] uses the representative training patches to explain a prediction in a self-interpretable framework for image classification. Koh et al. [2020] learn models that first predict human labeled concepts, then use concept scores to predict model. Goyal et al. [2019b] measures the causal effect of concepts by using a conditional VAE model. These methods either require a training

model from scratch or training a generative model, whereas our method can be applied on given models and different data types.

## 3 Defining Completeness of Concepts

**Problem setting:** Consider a set of $n$ training examples $\mathbf{x}^1, \mathbf{x}^2, ..., \mathbf{x}^n$, corresponding labels $y^1, y^2, ..., y^n$ and a given pre-trained DNN model that predicts the corresponding $y$ from the input $\mathbf{x}$. We assume that the pre-trained DNN model can be decomposed into two functions: the first part $\Phi(\cdot)$ maps input $\mathbf{x}^i$ into an intermediate layer $\Phi(\mathbf{x}^i)$, and the second part $h(\cdot)$ maps the intermediate layer $\Phi(\mathbf{x}^i)$ to the output $h(\Phi(\mathbf{x}^i))$, which is a probability vector for each class, and $h_y(\Phi(\mathbf{x}))$ is the probability of data $\mathbf{x}$ being predicted as label $y$ by the model $f$. For DNNs that build up by processing parts of input at a time, such as those composed of convolutional layers, we can additionally assume that $\Phi(\mathbf{x}^i)$ is the concatenation of $[\phi(\mathbf{x}_1^i), ..., \phi(\mathbf{x}_T^i)]$, such that $\Phi(\cdot) \in \mathbb{R}^{(T \cdot d)}$, and $\phi(\cdot) \in \mathbb{R}^d$. Here, $\mathbf{x}_1^i, \mathbf{x}_2^i, ..., \mathbf{x}_T^i$ denote different, potentially overlapping parts of the input for $\mathbf{x}^i$, such as a segment of an image or a sub-sentence of a text. These parts for example, can be chosen to correspond to the receptive field of the neurons at the intermediate layer $\Phi(\cdot)$. We will use these $\mathbf{x}_t^i$ to relate discovered concepts. As an illustration of such parts, consider the fifth convolution layer of a VGG-16 network with input shape $224 \times 224$ have the size $7 \times 7 \times 512$. If we treat this layer as $\Phi(\mathbf{x}^i)$, $\phi(\mathbf{x}_1^i)$ corresponds to the first 512 dimensions of the intermediate layer (with size $7 \times 7 \times 512$), and $\Phi(\mathbf{x}^i) = [\phi(\mathbf{x}_1^i), ..., \phi(\mathbf{x}_{49}^i)]$. Here, each $\mathbf{x}_j^i$ corresponds to a $164 \times 164$ square in the input image (with effective stride 16), which is the receptive field of convolution layer 5 of VGG-16 [Araujo et al., 2019]. We note that when the receptive field of $\phi(\cdot)$ is equal to the entire input size, such as for multi-layer perceptrons, we may simply choose $T = 1$ so that $\mathbf{x}_{1:T}^i = \mathbf{x}^i$ and $\Phi(\mathbf{x}^i) = \phi(\mathbf{x}_1^i)$. Thus, our method can also be generally applied to any DNN with an arbitrary structure besides convolutional layers. To choose the intermediate layer to apply concepts, we follow previous works on concept explanations Kim et al. [2018], Ghorbani et al. [2019] by starting from the layer closest to the prediction until we reached a layer that user is happy with, as higher layers encodes more abstract concepts with larger receptive field, and lower layers encodes more specific concepts with smaller receptive field.

Suppose that there is a set of $m$ concepts denoted by unit vectors[2] $\mathbf{c}_1, \mathbf{c}_2, ..., \mathbf{c}_m$ that represent linear directions in the activation space $\phi(\cdot) \in \mathbb{R}^d$, given by a concept discovery algorithm. For each part of data point $\mathbf{x}_t$ (We omit $i$ for notational simplicity), The inner product between the data and concept vector is viewed as the closeness of the input $\mathbf{x}_t$ and the concept $\mathbf{c}$ following Kim et al. [2018], Ghorbani et al. [2019]. If $\langle \phi(\mathbf{x}_t), c_j \rangle$ is large, then we know that $\mathbf{x}_t$ is close to concept $j$. However, when $\langle \phi(\mathbf{x}_t), c_j \rangle$ is less than some threshold, the dot product value is not semantically meaningful other than the the input is not close to the concept. Based on this motivation, we define the *concept product* for part of data $\mathbf{x}_t$ as $v_{\mathbf{c}}(\mathbf{x}_t) := \text{TH}(\langle \phi(\mathbf{x}_t), c_j \rangle, \beta)_{j=1}^m \in \mathbb{R}^m$, where TH is a threshold which trims value less than $\beta$ to 0. We normalize the concept product to unit norm for numerical stability, and aggregate upon all parts of data to obtain the *concept score* for input $\mathbf{x}$ as $v_{\mathbf{c}}(\mathbf{x}) = (\frac{v_c(x_t)}{\|v_c(x_t)\|_2})_{t=1}^T \in \mathbb{R}^{T \cdot m}$.

We assume that for "sufficient" concepts, the concept scores should be sufficient statistics for the model output, and thus we may evaluate the completeness of concepts by how well we can recover the prediction given the concept score. Let $g : \mathbb{R}^{T \cdot m} \to \mathbb{R}^{T \cdot d}$ denote any mapping from the concept score to the activation space of $\Phi(\cdot)$. If concept scores $v_{\mathbf{c}}(\cdot)$ are sufficient statistics for the model output, then there exists some mapping $g_f$ such that $h(g_f(v_{\mathbf{c}}(\mathbf{x}))) \approx f(\mathbf{x})$. We can now formally define the completeness core for a set of concept vectors $\mathbf{c}_1, ..., \mathbf{c}_m$:

**Definition 3.1. Completeness Score:** Given a prediction model $f(\mathbf{x}) = h(\phi(\mathbf{x}))$, a set of concept vectors $\mathbf{c}_1, ..., \mathbf{c}_m$, we define the completeness score $\eta_f(\mathbf{c}_1, ..., \mathbf{c}_m)$ as:

$$\eta_f(\mathbf{c}_1, ..., \mathbf{c}_m) = \frac{\sup_g \mathbb{P}_{\mathbf{x}, y \sim V}[y = \arg\max_{y'} h_{y'}(g(v_{\mathbf{c}}(\mathbf{x})))] - a_r}{\mathbb{P}_{\mathbf{x}, y \sim V}[y = \arg\max_{y'} f_{y'}(\mathbf{x})] - a_r}, \qquad (1)$$

where $V$ is the set of validation data and $\sup_g \mathbb{P}_{\mathbf{x}, y \sim V}[y = \arg\max_{y'} h_{y'}(g(v_{\mathbf{c}}(\mathbf{x})))]$ is the best accuracy by predicting the label just given the concept scores $v_{\mathbf{c}}(\mathbf{x})$, and $a_r$ is the accuracy of random prediction to equate the lower bound of completeness score to 0. When the target $y$ is multi-label, we may generalize the definition of completeness score by replacing the accuracy with the binary accuracy, which is the accuracy where each label is treated as a binary classification.

To calculate the completeness score, we can set $g$ to be a DNN or a simple linear projection, and optimize using stochastic gradient descent. In our experiments, we simply set $g$ to be a two-layer perceptron with 500 hidden units. We note that we approximate $f(\mathbf{x}_t)$ by $h(g_f(v_\mathbf{c}(\mathbf{x}_t)))$, but not an arbitrary neural network $h_g(v_\mathbf{c}(\mathbf{x}_t))$ for two benefits: (a) the measure of completeness considers the architecture and parameter of the given model to be explained (b) the computation is much more efficient since we only need to optimize the parameters of $g$, instead of the whole backbone $h_g$. The completeness score measures how "sufficient" are the concept scores as a sufficient statistic of the model, based on the assumption that the concept scores of "complete" concepts are sufficient statistics of the model prediction $f(\cdot)$. By measuring the accuracy achieved by the concept score, we are effectively measuring how "complete" the concepts are. We note that the completeness score can also be used to measure how sufficient concepts can explain a dataset independent of the model, by replacing $\phi(\cdot), h(\cdot)$ with identical functions, and $f(\mathbf{x})$ with $y$. Below is an illustrative example on why we need the completeness score:

**Example 3.1.** Consider a simplified scenario where we have the input $\mathbf{x} \in \mathbb{R}^m$, and the intermediate layer $\Phi$ is the identity function. In this case, the $m$ concepts $\mathbf{c}_1, \mathbf{c}_2, ..., \mathbf{c}_m$ are the one-hot encoding of each feature in $\mathbf{x}$. Assume that the concepts $\mathbf{c}_1, \mathbf{c}_2, ..., \mathbf{c}_m$ follow independent Bernoulli distribution with $p = 0.5$, and the model we attempt to explain is $f(\mathbf{x}) = \mathbf{c}_1 \text{ XOR } \mathbf{c}_2... \text{ XOR } \mathbf{c}_m$. The ground truth concepts that are sufficient to the model prediction should then be $\mathbf{c}_1, \mathbf{c}_2, ..., \mathbf{c}_m$. However, if we have the information on $\mathbf{c}_1, \mathbf{c}_2, ..., \mathbf{c}_{m-1}$ but do not have information on $\mathbf{c}_m$, we may have at most $0.5$ probability to predict the output of the model, which is the same as the accuracy of random guess. In this case, $\eta_f(\mathbf{c}_1, \mathbf{c}_2, ..., \mathbf{c}_{m-1}) = 0$. On the other hand, given $\mathbf{c}_1, \mathbf{c}_2, ..., \mathbf{c}_m$, $\eta_f(\mathbf{c}_1, \mathbf{c}_2, ..., \mathbf{c}_m) = 1$.

The completeness score offers a way to assess the 'sufficiency' of the discovered concepts to "explain" reasoning behind a model's decision. Not only the completeness score is useful in evaluating a proposed concept discovery method, but it can also shed light on how much of the learned information by DNN may not be 'understandable' to humans. For example, if the completeness score is very high, but discovered concepts aren't making cohesive sense to humans, this may mean that the DNN is basing its decisions on other concepts that are potentially hard to explain.

## 4 Discovering Completeness-aware Interpretable Concepts

### 4.1 Limitations of existing methods

Our goal is to discover a set of maximally-complete concepts under the definition 4.2, where each concept is interpretable and semantically-meaningful to humans. We first discuss the limitations of recent notable works related to concept discovery and then explain how we address them. TCAV and ACE are concept discovery methods that use training data for specific concepts and use trained linear concept classifier to derive concept vectors. They quantify the saliency of a concept to a class using 'TCAV score', based on the similarity of the loss gradients to the concept vectors. This score implicitly assumes a first-order relationship between the concepts and the model outputs. Regarding labeling of the concepts, TCAV relies on human-defined labels, while ACE uses automatically-derived image clusters by k-means clustering of super-pixel segmentations. There are two main caveats to these approaches. The first is that while they may retrieve an important set of concepts, there is no guarantee on how 'complete' the concepts are in explain the model – e.g., one may have 10 concepts with high TCAV scores, but they may still be very insufficient in understanding the predictions. Besides, human-suggested exogenous concept data might even encode confirmation bias. The second caveat is that their saliency scores may fail to capture concepts that have non-linear relationships with the output due to first-order assumption. The concepts in Example 3.1 might not be retrieved by the TCAV score since XOR is not a linear relationship. Overall, our completeness score complements previous works in concept discovery by adding a criterion to determine whether a set of concepts are sufficient to explain the model. The discussion of our relation to PCA is in the Appendix.

### 4.2 Our method

The goal of our method is to obtain concepts that are *complete* to the model. We consider the case where each data point $\mathbf{x}^i$ has parts $\mathbf{x}^i_{1:T}$, as described above. We assume that input data has spatial dependency, which can help learning coherent concepts. Thus, we encourage proximity between each concept and its nearest neighbors patches. Note that the assumption works well with images and language, as we will demonstrate in the result section. We aim that the concepts would obtain consistent nearest neighbors that only occur in parts of the input, e.g. head of animals or the grass in the background so that the concepts are pertained to certain spacial regions. By encouraging

the closeness between each concept and its nearest neighbors, we aim to obtain consistent nearest neighbors to enhance interpretability. Lastly, we optimize the completeness terms to encourage the *completeness* of the discovered concepts.

**Learning concepts:** To optimize the completeness of the discovered concepts, we optimize the surrogate loss for the completeness term for both concept vectors $\mathbf{c}_{1:m}$ and the mapping function $g$:

$$\underset{\mathbf{c}_{1:m}, g}{\arg\max} \log \mathbb{P}[h_y(g(v_{\mathbf{c}}(\mathbf{x})))] \tag{2}$$

An interpretation for finding the underlying concepts whose concept score maximizes the recovered prediction score is analogous to treating the prediction of DNNs as a topic model. By assuming the data generation process of $(\mathbf{x}, y)$ follows the probabilistic graphical model $\mathbf{x}_t \to \mathbf{z}_t$ and $\mathbf{z}_{1:T} \to y$, such that the concept assignment $\mathbf{z}_t$ is generated by the data, and the overall concept assignment $\mathbf{z}_{1:T}$ determines the label $y$. The log likelihood of the data $\log P[y|\mathbf{x}]$ can be estimated by $\log P[y|\mathbf{x}] = \log \int_z P[y|z]P[z|\mathbf{x}] \approx \log P[y|E[z|\mathbf{x}]]$, by replacing the sampling by a deterministic average. We note that $v_{\mathbf{c}}(\mathbf{x}_{1:T})$ resembles $E[z|\mathbf{x}]$ and $P(y|h(g(v_{\mathbf{c}}(\mathbf{x})))$ resembles $P[y|E[z|\mathbf{x}]]$, and as in supervised topic modeling Mcauliffe and Blei [2008], we jointly optimize the latent "topic" and the prediction model, but in an end-to-end fashion to maintain efficiency instead of EM update.

To enhance the interpretability of our concepts beyond "topics", we further design a regularizer to encourage the spacial dependency (and thus coherency) of concepts. Intuitively, we require that the top-K nearest neighbor training input patches of each concept to be sufficiently close to the concept, and different concepts are as different as possible. This formulation encourages the top-K nearest neighbors of the concepts would be coherent, and thus allows explainability by ostensive definition. K is a hyperparameter that is usually chosen based on domain knowledge of the desired frequency of concepts. In our results, we fix K to be half of the average class size in our experiments. When using batch update, we find that picking $K = (\text{batch size} \cdot \text{average class ratio})/2$ works well in our experiments, where average class ratio = average instance of each class/total number of instances. That is, the regularizer term tries to maximize $\Phi(\mathbf{x}_t^i) \cdot \mathbf{c}_k$ while minimizing $\mathbf{c}_j \cdot \mathbf{c}_k$. $\Phi(\mathbf{x}_t^i) \cdot \mathbf{c}_k$ is the similarity between the $t^{th}$ patch of the $i^{th}$ example and $\mathbf{c}_j \cdot \mathbf{c}_k$ is the similarity between the $j^{th}$ concept vector and the $k^{th}$ concept vector. By averaging over all concepts, and defining $T_{\mathbf{c}_k}$ as the set of top-K nearest neighbors of $\mathbf{c}_k$, the final regularization term is

$$R(\mathbf{c}) = \lambda_1 \frac{\sum_{k=1}^m \sum_{\mathbf{x}_a^b \in T_{\mathbf{c}_k}} \Phi(\mathbf{x}_a^b) \cdot \mathbf{c}_k}{mK} - \lambda_2 \frac{\sum_{j \neq k} \mathbf{c}_j \cdot \mathbf{c}_k}{m(m-1)}.$$

By adding the regularization term to (2), the final objective becomes

$$\underset{\mathbf{c}_{1:m}, g}{\arg\max} \log P(h_y(g(v_{\mathbf{c}}(\mathbf{x}_{1:T})))) + R(\mathbf{c}), \tag{3}$$

for which we use stochastic gradient descent to optimize variables $\mathbf{c}_{1:m}, g$ jointly. When the optimization converges, $g$ is a (local) optimal value given $\mathbf{c}_{1:m}$. Since only concept vectors $\mathbf{c}_{1:m}$, and the mapping function $g$ is optimized in the process, the optimization process converges much faster compared to training the model from scratch. The computational cost for discovering concepts and calculating conceptSHAP is about 3 hours for AwA dataset and less than 20 minutes for the toy dataset and IMDB, using a single 1080 Ti GPU, which can be further accelerated with parallelism. The choice of which layer to apply $h_y, g$ and the corresponding architecture are further discussed in the appendix.

### 4.3 ConceptSHAP: How important is each concept?

Given a set of concept vectors $C_S = \{\mathbf{c}_1, \mathbf{c}_2, ...\mathbf{c}_m\}$ with a high completeness score, we would like to evaluate the importance of each individual concept by quantifying how much each individual concept contributes to the final completeness score. Let $\mathbf{s}_i$ denote the importance score for concept $\mathbf{c}_i$, such that $\mathbf{s}_i$ quantifies how much of the completeness score $\eta(C_S)$ is contributed by $\mathbf{c}_i$. Motivated by its successful applications in quantifying attributes for complex systems, we adapt Shapley values [Shapley, 1988] to fairly assign the importance of each concept (which we call ConceptSHAP):

**Definition 4.1.** Given a set of concepts $C_S = \{\mathbf{c}_1, \mathbf{c}_2, ...\mathbf{c}_m\}$ and some completeness score $\eta$, we define the ConceptSHAP $\mathbf{s}_i$ for concept $\mathbf{c}_i$ as

$$\mathbf{s}_i(\eta) = \sum_{S \subseteq C_s \backslash \mathbf{c}_i} \frac{(m - |S| - 1)!|S|!}{m!} [\eta(S \cup \{\mathbf{c}_i\}) - \eta(S)],$$

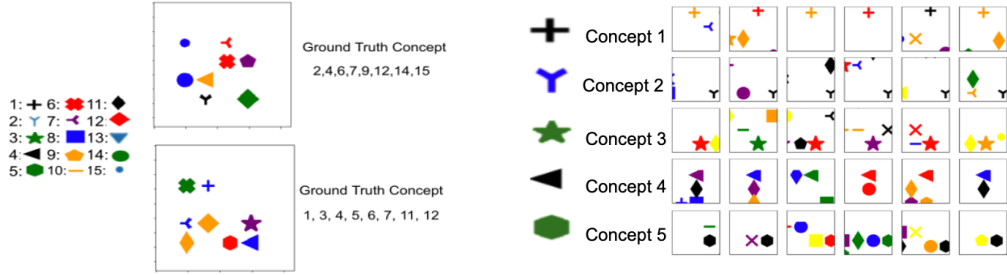

(a) Two random images and corresponding ground truth concepts (with their legend on the left) – each object corresponds to a ground truth concept solely via the shape information.

(b) Top nearest neighbors (each neighbor corresponds to a part of the full image) of each discovered concepts. The ground truth concepts, determined by their shape (with random colors), are on the left.

Figure 1: Examples (left) and nearest neighbors of our method (right) on Synthetic data.

The main benefit of Shapley for importance scoring is that it uniquely satisfies the set of desired axioms: efficiency, symmetry, dummy, and additivity. As these axioms are widely discussed in previous works [Shapley, 1988, Lundberg and Lee, 2017], we leave the definitions and proof to Appendix.

**Per-class saliency of concepts:** Thus far, conceptSHAP measures the global attribution (i.e., contribution to completeness when all classes are considered). However, per-class saliency, how much concepts contribute to prediction of a particular class, might be informative in many cases. To obtain the concept importance score for each class, we define the completeness score with respect to the class by considering data points that belong to it, which is formalized as:

**Definition 4.2.** Given a prediction model $f(\mathbf{x}) = h(\phi(\mathbf{x}))$, a set of concept vectors $\mathbf{c}_1, \mathbf{c}_2, ..., \mathbf{c}_m$ that lie in the feature subspace in $\phi(\cdot)$, we define the completeness score $\eta_{f,j}(\mathbf{c}_1, ..., \mathbf{c}_m)$ for class $j$ as:

$$\eta_{f,j}(\mathbf{c}_1, ..., \mathbf{c}_m) = \frac{\mathbb{P}_{\mathbf{x}, y \in V_j}[y = \arg\max_{y'} h_{y'}(\hat{g}(v_{\mathbf{c}}(\mathbf{x})))] - a_{r,j}}{\mathbb{P}_{\mathbf{x}, y \in V}[y = \arg\max_{y'} f_{y'}(\mathbf{x}_{1:T})] - a_r}, \quad (4)$$

where $V_j$ is the set of validation data with ground truth label $j$, and $a_{r,j}$ is the accuracy of random predictions for data in class $j$, and $\hat{g}$ is derived via the optimization of completeness. We then define the perclass ConceptSHAP for concept $i$ with respect to class $j$ as:

**Definition 4.3.** Given a prediction model $f(\mathbf{x})$, a set of concept vectors in the feature subspace in $\phi(\cdot)$. We can define the perclass ConceptSHAP for concept $i$ with respect to class $j$ as: $\mathbf{s}_{i,j}(\eta) = \mathbf{s}_i(\eta_j)$.

For each class $j$, we may select the concepts with the highest conceptSHAP score with respect to class $j$. We note that $\sum_j \frac{|V_j|}{|V|}\eta_j = \eta$ and thus with the additivity axiom, $\sum_j \frac{|V_j|}{|V|}\mathbf{s}_{i,j}(\eta_j) = \mathbf{s}_i(\eta)$.

# 5 Experiments

In this section, we demonstrate our method both on a synthetic dataset, where we have ground truth concept importance, as well as on real-world image and language datasets.

## 5.1 Synthetic data with ground truth concepts

**Setting:** We construct a synthetic image dataset with known and complete concepts, to evaluate how accurately the proposed concept discovery algorithm can extract them. In this dataset, each image contains at most 15 shapes (shown in Fig. 1a), and only 5 of them are relevant for the ground truth class, by construction. For each sample $\mathbf{x}^i$, $\mathbf{z}_j^i$ is a binary variable which represents whether $\mathbf{x}^i$ contains shape $j$. $\mathbf{z}_{1:15}^i$ is a 15-dimensional binary variable with elements independently sampled from Bernoulli distribution with $p = 0.5$. We construct a 15-dimensional multi-label target for each sample, where the target of sample $i$, $y^i$ is a function that depends only on $\mathbf{z}_{1:5}^i$, which represents whether the first 5 shape exists in $\mathbf{x}^i$. For example, $y_1 = \sim (\mathbf{z_1} \cdot \mathbf{z_3}) + \mathbf{z_4}, y_2 = \mathbf{z_2} + \mathbf{z_3} + \mathbf{z_4}, y_3 = \mathbf{z_2} \cdot \mathbf{z_3} + \mathbf{z_4} \cdot \mathbf{z_5}$, where $\sim$ denotes logical Not (details are in Appendix). We construct 48k training samples and 12k evaluation samples and use a convolutional neural network with 5 layers, obtaining 0.999 accuracy. We take the last convolution layer as the feature layer $\phi(\mathbf{x})$.

Table 1: The average number of correct and agreed concepts by users based on nearest neighbors.

|  | ACE | ACE-SP | PCA | k-means | **Ours** |
|---|---|---|---|---|---|
| correct concepts | $3.0 \pm 0$ | $2.75 \pm 0.46$ | $3.875 \pm 0.35$ | $3.75 \pm 0.46$ | $\mathbf{5.0 \pm 0}$ |
| agreed concepts | 4.625 | 4.75 | 4.375 | 4.75 | **5.0** |
| automated alignment | 0.741 | − | 0.876 | 0.864 | **0.98** |

**Evaluations:** We conduct a user-study with 20 users to evaluate the nearest neighbor samples of a few concept discovery methods. At each question, a user sees 10 nearest neighbor images of each discovered concept vector (as shown on the right of Fig. 1b), and is asked to choose the most common and coherent shape out of the 15 shapes based on the 10 nearest neighbors. We evaluate the results for our method, k-means clustering, PCA, ACE, and ACE-SP when $m = 5$ concepts are retrieved. Each user is tested on two randomly chosen methods in random order, and thus each method is tested on 8 users. We report the average number of correct concepts and the number of agreed concepts (where the mode of each question is chosen as the correct answer) for each method answered by users in Table 1. The average number of correct concepts measures how many of the correct concepts are retrieved by user via nearest neighbors. The average number of agreed concepts measures how consistent are the shapes retrieved by different users, which is related to the coherency and conciseness of the nearest neighbors for each method. We also provide an automated alignment score based on how the discovered concept direction classifies different concepts – see Appendix for details.

**Results:** We compare our methods to ACE, k-means clustering, and PCA. For k-means and PCA, we take the embedding of the patch as input to be consistent to our method. For ACE, we implement a version which replaces the superpixels with patches and another version that takes superpixels as input, which we refer as ACE and ACE-SP respectively. We report the correct concepts and agreed concepts from the user study, and an automated alignment score which does not require humans. We do not calculate the alignment score of ACE-SP since it does not operate on patches and thus is unfair to compare with others (which would lead to much lower scores.) Our method outperforms others on corrected concepts and alignment score, is superior in retrieving the accurate concepts beyond the limitations of others. The number of agreed concepts is also the highest for our method, showing how highly-interpretability it is to humans such that the same concepts are consistently retrieved based on nearest neighbors. As qualitative results, Fig. 1b shows the top-6 nearest neighbors for each concept $\mathbf{c}_k$ of our concept discovery method based on the dot product $\langle \mathbf{c}_k, \Phi(\mathbf{x}_a)^b \rangle$. All nearest neighbors contain a specific shape that corresponds to the ground-truth shapes 1 to 5. For example, all nearest neighbors of concept 1 contain the ground truth shape 1, which are cross as listed in Fig. 1a. A complete list of the top-10 nearest neighbors of all concept discovery methods is shown in Appendix.

## 5.2 Image classification

**Setting and metrics:** We perform experiments on Animals with Attribute (AwA) [Lampert et al., 2009] that contains 50 animal classes. We use 26905 images for training and 2965 images for evaluation. We use the Inception-V3 model, pre-trained on Imagenet [Szegedy et al., 2016], which yields 0.9 test accuracy. We apply our concept discovery algorithm to obtain $m = 70$ concepts. We conduct ad-hoc duplicate concept removal, by removing one concept vector if there are two vectors where the dot product is over 0.95. This gives us 53 concepts in total. We then calculate the ConceptSHAP and per class saliency score for each concept and each class. For each class, the top concepts based on the conceptSHAP are the most important concepts to classify this class, as

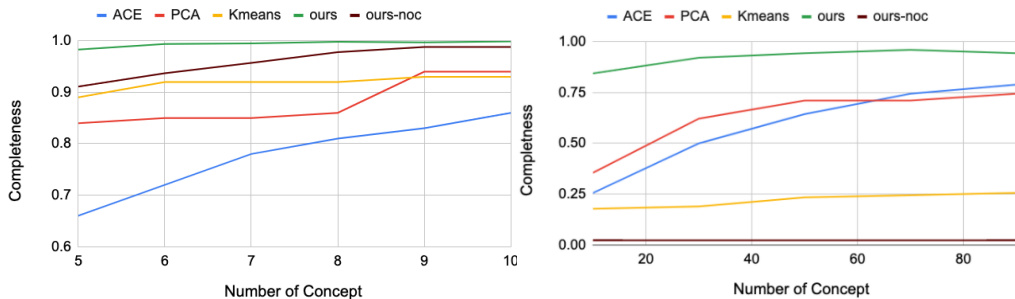

Figure 2: Completeness scores on synthetic dataset (left) and completeness scores on AwA (right) versus different number of discovered concepts $m$ for all concept discovery methods in the synthetic dataset. Ours-noc refers to our method without the completeness score objective as an ablation study.

shown in Fig.3. While ConceptSHAP is useful in capturing the sufficiency of concepts for prediction, sometimes we may want to show examples. We propose to measure the quality of the nearest neighbors explanations by the average dot product between the nearest-neighbor patches that belongs to the class and the concept vector. In other words, the quality of the nearest neighbors explanations is simply the first term in $R(\mathbf{c})$, which we denote as $R_1(\mathbf{c}) = \sum_{k=1}^{m} \sum_{\mathbf{x}_a^b \subseteq T_{\mathbf{c}_k}} \langle \Phi(\mathbf{x}_a^b), \mathbf{c}_k \rangle$, where the top-K set is limited to image patches in the class of interest. When the nearest neighbor set contains patches of the same original image, we only show the patch with the highest similarity to the concept to increase the diversity.

**Results:** We show the top concepts (ranked by conceptSHAP) of 3 classes with $R_1(\mathbf{c})$>0.8 in Fig. 3 (full results are in Appendix). Note that since our method finds concepts for *all* classes as opposed to specific to one class (such as Ghorbani et al. [2019], Chen et al. [2019]), we discover common concepts across many classes. For example, concept 7, whose nearest neighbors show grass texture, is important for the classes 'Squirrel', 'Rabbit', 'Bob Cat', since all these animals appear in prairie. Concept 8 shows a oval head and large black round eyes shared by the classes 'Rabbit', 'Squirrel', 'Weasel', while concept 46 shows head of the 'Bob cat', which is shared by the classes 'Lion', 'Leopard', and 'Tiger', 'Antelope', and 'Gorilla', which all show animal heads that are more rectangular and significantly different to the animal heads of concept 8. We find that having concepts shared between classes is useful to interpret the model. Fig. 2 shows that our method achieves the highest completeness of all methods on both the synthetic dataset and AwA. As a sanity check, we include the baseline 'ours-noc', where the completeness objective is removed from (3). Our method has much higher completeness than 'ours-noc', demonstrating the necessity of the completeness term. In Appendix, we show some more top concepts for PCA, Kmeans, where the top concepts for PCA and Kmeans are also chosen as the same setting as ours.

**Human Study:** We conduct a human study for the top neighbors for concepts discovered by our method, PCA, and Kmeans on the classes 'Squirrel', 'Rabbit', 'Bob Cat'. For each method, we randomly choose 1 top concept per class for the 3 classes, and thus we choose 3 concepts per method (9 concepts in total). For each concept, we show users 4 top images of that concept, and ask users to choose the image (out of 3 different options) that they believe should belong to the same concept (where one of the option will actually belong to the same concept, and the other two are random image patches of the same class that does not belong to that concept). We then calculate the average accuracy to measure the human interpretability of the concept discover method. We conduct a user study with 10 users, where each of them are asked with the same 9 questions (1 question per concept chosen). The average correct ratio for our method, PCA, and Kmeans are 0.733, 0.267, and 0.6 respectively, showing our method's superiority. Kmeans outperforms PCA as it also encourages closeness of top nearest neighbors (which is better for ostensive definition).

### 5.3 Text classification

**Setting:** We apply our method on IMDB, a text dataset with movie reviews classified as either positive or negative. We use 37500 reviews for training and 12500 for testing. We employ a 4-layer CNN model with 0.9 test accuracy. We apply our concept discover method to obtain 4 concepts, where the part of data $\mathbf{x}_j^i$ consists of 10 consecutive words of the sentence. The completeness of the 4 concepts is 0.97, thus the 4 concepts are highly representative of the classification model.

**Result:** For each concept, Table 2 shows (a) the top nearest neighbors based on the dot product of the concept and part of reviews (b) the most frequent words in the top-500 nearest neighbors

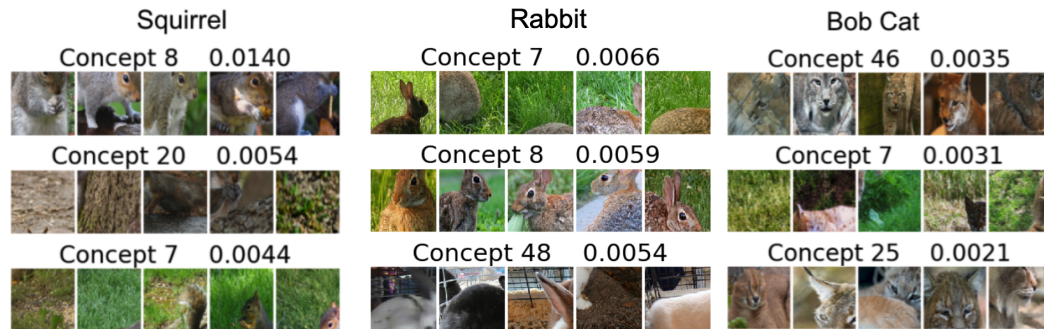

Figure 3: Concept examples with the samples that are the nearest to concept vectors in the activation space in AwA. The per-class ConceptSHAP score is listed above the images.

Table 2: The 4 discovered concepts and some nearest neighbors along with the most frequent words that appear in top-500 nearest neighbors.

| Concept | Nearest Neighbors | Frequent words | ConceptSHAP |
|---|---|---|---|
| 1 | poorly constructed what comes across as interesting is the wasting my time with a comment but this movie awful in my opinion there were \<UNK\> and the | worst (168) ever (69) movie (61) seen (55) film (50) awful (42) time(40) waste (34) poorly (26) movies (24) films (18) long (17) | 0.280 |
| 2 | normally it would earn at least 2 or 3 \<UNK\> \<UNK\> is just too dumb to be called i feel like i was ripped off and hollywood | not (58) movie (39) make (25) too (23) film (22) even (19) like (18) 2 (16) never (14) minutes (13) 1 (12) doesn't (11) | 0.306 |
| 3 | remember awaiting return of the jedi with almost \<UNK\> better than most sequels for tv movies i hate male because marie has a crush on her attractive | movies (19) like (18) see (16) movie (15) love (15) good (12) character (11) life (11) little (10) ever (9) watch (9) first (9) | 0.174 |
| 4 | new \<UNK\> \<UNK\> via \<UNK\> \<UNK\> with absolutely hilarious homosexual and an italian clown \<UNK\> is an entertaining stephen \<UNK\> on the vampire \<UNK\> as a masterpiece | excellent (50) film (25) perfectly (19) wonderful (19) perfect (16) hilarious (15) best (13) fun (12) highly (11) movie (11) brilliant (9) old (9) | 0.141 |

(excluding stop words) (c) the conceptSHAP score for each concept. We can see that concepts 1 and 2 contain mostly negative sentiments, evident from the nearest neighbors – concept 1 tends to criticize the movie/film directly, while concept 2 contains negativity in comments via words such as "not", "doesn't", "even". We note that the ratings in concept 2 are also negative since the scores 1 and 2 are considered to be very negative in movie review. On the other hand, concepts 3 and 4 contain mostly positive sentiments, as evident from the nearest neighbors – concept 3 seems to discuss the plot of the movie without directing acclaiming or criticizing the movie, while concept 4 often contains very positive adjectives such as "excellent", "wonderful" that are extremely positive. More nearest neighbors are provided in the Appendix.

**Appending discovered concepts:** We perform an additional experiment where we randomly append 5 nearest neighbors (out of 500-nearest neighbors) of each concept to the end of all testing instances for further validation of the usefulness of the discovered concepts. For example, we may add "wasting my time with a comment but this movie" along with 4 other nearest neighbors of concept 1 to the end of a testing sentence. The original average prediction score for the testing sentences is 0.516, and the average prediction score after randomly appending 5 nearest neighbors of each concept becomes 0.103, 0.364, 0.594, 0.678 for concept 1, 2, 3, 4. As a controlled experiment, we appended 5 random sentences to the testing sentences, and the average prediction score is 0.498. This suggests that the concept score is highly related to the how the model makes prediction and may be used to manipulate the prediction. We note that while concept 1 contains stronger and more direct negative words than concept 2, concept 2 has a higher conceptSHAP value than concept 1. We hypothesize this is due to the fact that concept 2 may better detect weak negative sentences that may be difficult to be explained by concept 1, and thus may contribute more to the completeness score.

## 6 Conclusions

We propose to quantify the sufficiency of a particular set of concepts in explaining the model's behavior by the *completeness* of concepts. By optimizing the completeness term coupled with additional constraints to ensure interpretability, we can discover concepts that are complete and interpretable. Through experiments on synthetic and real-world image and language data, we demonstrate that our method can recover ground truth concepts correctly, and provide conceptual insights of the model by examining the nearest neighbors. Although our work focuses on post-hoc explainability of pre-trained DNNs, joint training with our proposed objective function is possible to train inherently-interpretable DNNs. An interesting future direction is exploring the benefits of joint learning of the concepts along with the model, for better interpretability.

# 7 Broader Impact

Bringing explainability can be crucial for AI deployments, for decision makers to build trust, for users to understand decisions, and for model developers to improve the quality. There are many use cases, from Finance, Healthcare, Employment/Recruiting, Retail, Environmental Sciences etc., that the explainability indeed constitutes the bottleneck to use deep neural networks (DNNs) despite their high performance. Thus, bringing explainability to DNNs can open many horizons for new AI deployments.

There are different forms of explainability, and our contributions are specifically for the very important 'concept-based' explanations, towards a coherent and complete transparency to DNNs. Our paper contributes to the quantification of the "completeness" of concept explanations, which can be useful to evaluate how sufficient existing and future concept-based explanations are, on the task of explaining a neural network. Validating the how sufficient the explanations are in explaining the model is a necessary sanity check, but often overlooked for concept explanations. As the field of explainable AI progresses rapidly, critiques and doubts on whether explanations are actually useful and accountable for models have also increased. Our objective metric bridges the gap between existing explanations and model accountability.

Most post-hoc concept-based explanations are applied only on image data. Our method is data type agnostic. We demonstrate our canonical idea on both image and language data, which we believe can be applied to other data types as well. We believe our work lays a groundwork on general concept-based explanations, and hopefully will encourage future works on exploring concept explanations on all kinds of data types. Our concept discovery method explain a model using a small number of concepts, which can be explained by providing nearest neighbors in the training data to help users understand the concepts better. Such explanations provide a broader understanding of the model compared to the widely-used methods, such as saliency maps, and can be helpful for model developers and data scientists in understanding how the model works, improving the model based on insights, and ultimately learning from AI systems to build better AI systems.

# 8 Ackowlegement

We acknowledge the support of DARPA via FA87501720152.

## Footnotes

[1]The code is released at https://github.com/chihkuanyeh/concept_exp.

[2] We apply additional normalization to $\phi(\cdot)$ so it has unit norm and keep the notation for simplicity.

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
