[Supplementary Material]

# Appendix A    Relation to PCA

**PCA:**    We show that under strict conditions, the PCA vectors applied on an intermediate layer where the principle components are used as concept vectors, maximizes the completeness score.

**Proposition A.1.** *When $h$ is an isometry function that maps from $(\Phi(\cdot), \|\cdot\|_F) \rightarrow (f(\cdot), \|\cdot\|_F)$, and additionally $f(\mathbf{x}_i) = \mathbb{1}[y_i]$, $\forall (\mathbf{x}_i, y_i) \in V$ (i.e. the loss is minimized, $\mathbb{1}[y_i]$ is the one hot vector of class $y_i$), and also assume $T = 1$, $\mathbb{E}[z] = \langle \phi(\mathbf{x}), \mathbf{c} \rangle$, and $l$ is a linear function, the first $m$ PCA vectors maximizes the L2 surrogate of $\eta$.*

We note that the assumptions for this proposition are extremely stringent, and may not hold in general. When the isometry and other assumptions do not hold, PCA no longer maximizes the completeness score as the lowest reconstruction in the intermediate layer do not imply the highest prediction accuracy in the output. In fact, DNNs are shown to be very sensitive to small perturbations in the input [Narodytska and Kasiviswanathan, 2017] – they can yield very different outputs although the difference in the input is small (and often perceptually hard to recognize to humans). Thus, even though the reconstruction loss between two inputs are low at an intermediate layer, subsequent deep nonlinear processing may cause them to diverge significantly. The principal components are also not trained to be semantically meaningful, but to greedily minimize the reconstruction error (or maximize the projected variance). Even though completeness score and PCA share the idea of minimizing the reconstruction loss via dimensionality reduction, the lack of human interpretability of the principle components is a major bottleneck for PCA.

We provide this proposition only because completeness and PCA share the idea of minimizing the reconstruction loss via dimensionality reduction. Another notable limitation of using PCA as concept vectors is the lack of human interpretability of the principle components. The PCA vectors are not trained to be semantically meaningful, but to greedily minimize the reconstruction error (or maximize the projected variance).

**Proof of Proposition A.1**

*Proof.*  By the basic properties of PCA, the first $m$ PCA vectors (principal components) minimize the reconstruction $\ell_2$ error. Define the concatenation of the $m$ PCA vectors as a matrix $\mathbf{p}$ and $\|\cdot\|$ as the $\ell_2$ norm, and define $\mathrm{proj}(\phi(\mathbf{x}, \mathbf{p}))$ as the projection of $\mathbf{x}$ onto the span of $\mathbf{p}$, the basic properties of PCA is equivalent to that for all $\mathbf{c} = [\mathbf{c}_1 \ \mathbf{c}_2 \ldots \ \mathbf{c}_m]$,

$$\sum_{\mathbf{x} \subseteq V_X} \|\mathrm{proj}(\phi(\mathbf{x}), \mathbf{p}) - \phi(\mathbf{x})\|_F^2 \leqslant \sum_{\mathbf{x} \subseteq V_X} \|\mathrm{proj}(\phi(\mathbf{x}), \mathbf{c}) - \phi(\mathbf{x})\|_F^2.$$

By the isometry of $h$, we have

$$\sum_{\mathbf{x} \subseteq V_X} \|h(\mathrm{proj}(\phi(\mathbf{x}), \mathbf{p})) - h(\phi(\mathbf{x}))\|_F^2 \leqslant \sum_{\mathbf{x} \subseteq V_X} \|h(\mathrm{proj}(\phi(\mathbf{x}), \mathbf{c})) - h(\phi(\mathbf{x}))\|_F^2,$$

and since $f(\mathbf{x})$ is equal to Y, we can rewrite to

$$\sum_{\mathbf{x}, y \subseteq V} \|h(\mathrm{proj}(\phi(\mathbf{x}), \mathbf{p})) - \mathbb{1}[y]\|_F^2 \leqslant \sum_{\mathbf{x}, y \subseteq V} \|h(\mathrm{proj}(\phi(\mathbf{x}), \mathbf{c})) - \mathbb{1}[y]\|_F^2. \tag{5}$$

We note that under the assumptions, $\mathbb{E}[\mathbf{z}|\mathbf{x}] = \phi(\mathbf{x})\mathbf{c}$, and thus the reconstruction layer $l$ can be written as

$$
\begin{aligned}
l &= \arg\max_{l} \sum_{\mathbf{x}, y \subseteq V} \|\mathbb{1}[y] - h(l(\mathbb{E}[\mathbf{z}|\mathbf{x}]))\|_F^2 \\
&= \arg\max_{l} \sum_{\mathbf{x}, y \subseteq V} \|\mathbb{1}[y] - h(l(\phi(\mathbf{x})\mathbf{c}))\|_F^2 \\
&= \arg\max_{l} \sum_{\mathbf{x} \subseteq V_x} \|\phi(\mathbf{x}) - l(\phi(\mathbf{x})\mathbf{c})\|_F^2,
\end{aligned}
\tag{6}
$$

By definition, $\sum_{\mathbf{x} \subseteq V_x} \|\phi(\mathbf{x}) - l(\phi(\mathbf{x})\mathbf{c})\|_F^2$ is minimized by the projection, and thus $l(\phi(\mathbf{x})\mathbf{c}) = \mathrm{proj}(\phi(\mathbf{x}), \mathbf{c})$.

And thus, (5) can be written as:

$$\sum_{\mathbf{x},y \subseteq V} \|h(l(\phi(\mathbf{x})\mathbf{p})) - \mathbb{1}[y]\|_F^2 \leqslant \sum_{\mathbf{x},y \subseteq V} \|h(l(\phi(\mathbf{x})\mathbf{c})) - \mathbb{1}[y]\|_F^2.$$

and subsequently get that for any $\mathbf{c}$

$$\frac{\mathbb{E}_{\mathbf{x},y \sim V}[\|\mathbb{1}[y] - P(y'|\mathbb{E}[z_{1:T}], h, \mathbf{p})\|_F^2] - R}{\mathbb{E}_{\mathbf{x},y \sim V}[\|\mathbb{1}[y] - P(\mathbf{x}_{1:T}, f)\|_F^2] - R} \geqslant \frac{\mathbb{E}_{\mathbf{x},y \sim V}[\|\mathbb{1}[y] - P(y'|\mathbb{E}[z_{1:T}], h, \mathbf{c})\|_F^2] - R}{\mathbb{E}_{\mathbf{x},y \sim V}[\|\mathbb{1}[y] - P(\mathbf{x}_{1:T}, f)\|_F^2] - R}.$$

$\square$

Thus, PCA vectors maximize the L2 surrogate of the completeness score. We emphasize that Proposition A.1 has several assumptions that may not be practical. However, the proposition is only meant to show that PCA optimizes our definition of completeness under a very stringent condition, as the key idea of completeness and PCA are both to prevent information loss through dimension reduction.

## Appendix B    Shapley Axioms for ConceptSHAP

The axiomatic properties for ConceptSHAP are listed in the following proposition:

**Proposition B.1.** *Given a set of concepts $C_S = \{\mathbf{c}_1, \mathbf{c}_2, ...\mathbf{c}_m\}$ and a completeness score $\eta$, and some importance score $\mathbf{s}_i$ for each concept $\mathbf{c}_i$ that depends on the completeness score $\eta$. $\mathbf{s}_i$ defined by conceptSHAP is the unique importance assignment that satisfy the following four axioms:*

- *Efficiency: The sum of all importance value should sum up to the total completeness score, $\sum_{i=1}^m \mathbf{s}_i(\eta) = \eta(C_S)$.*

- *Symmetry: For two concept that are equivalent s.t. $\eta(u \cup \{\mathbf{c}_i\}) = \eta(u \cup \{\mathbf{c}_j\})$ for every subset $u \subseteq C_S \backslash \{\mathbf{c}_i, \mathbf{c}_j\}$, $\mathbf{s}_i(\eta) = \mathbf{s}_j(\eta)$.*

- *Dummy: If $\eta(u \cup \{\mathbf{c}_i\}) = \eta(u)$ for every subset $u \subseteq C_S \backslash \{\mathbf{c}_i\}$, then $\mathbf{s}_i(\eta) = 0$.*

- *Additivity: If $\eta$ and $\eta'$ have importance value $\mathbf{s}(\eta)$ and $\mathbf{s}(\eta')$ respectively, then the importance value of the sum of two completeness score should be equal to the sum of the two importance values, i.e, $\mathbf{s}_i(\eta + \eta') = \mathbf{s}_i(\eta) + \mathbf{s}_i(\eta')$ for all i.*

The proof and the interpretation for these concepts are well discussed in [Shapley, 1988, Lundberg and Lee, 2017, Fujimoto et al., 2006].

## Appendix C    Additional Experiments Results and Settings

**Automated Alignment score on Synthetic Dataset**    Given the existence of each ground truth shape $\mathbf{z}_{1:5}^i$ in each sample $\mathbf{x}^i$, we can evaluate how closely the discovered concept vectors $\mathbf{c}_{1:m}$ align with the actual ground truth shapes 1 to 5. Our evaluation assumes that if $\mathbf{c}_k$ corresponds to some shape $v$, then the parts of input that contain the shape $v$ and the parts of input that does not contain ground truth shape $v$ can be linearly separated by $\mathbf{c}_k$. That is, $\mathbf{c}_k \cdot \mathbf{x}_a > \mathbf{c}_k \cdot \mathbf{x}_b$ or $\mathbf{c}_k \cdot \mathbf{x}_a < \mathbf{c}_k \cdot \mathbf{x}_b$ for all $\mathbf{x}_a$ that contains shape $v$ and all $\mathbf{x}_b$ that does not contain shape $v$. Without loss of generality, we assume $\mathbf{c}_k \cdot \mathbf{x}_a > \mathbf{c}_k \cdot \mathbf{x}_b$ if $\mathbf{x}_a$ contains shape $v$ and $\mathbf{x}_c^d$ does not contain shape $v$ for notation simplicity, and check $\mathbf{c}_k$ and $-\mathbf{c}_k$ for each discovered concepts. Following this assumption, $\max_{t=1}^T \mathbf{c}_k \cdot \mathbf{x}_t^i > \max_{t=1}^T \mathbf{c}_k \cdot \mathbf{x}_t^j$ for all $i, j$ such that $\mathbf{z}_v^i = 1$ and $\mathbf{z}_v^j = 0$, since at least one part of $\mathbf{x}_t^b$ should contain the ground truth shape $v$. Therefore, to evaluate how well $\mathbf{c}_k$ corresponds to shape $v$, we measure the accuracy of using $\mathbb{1}[\max_{t=1}^T \mathbf{c}_k \cdot \mathbf{x}_t^i > const]$ to classify $\mathbf{z}_v^i$. More formally, we define the matching score between concept $\mathbf{c}_k$ to the shape v as:

$$\text{Match}(\mathbf{c}_k, \mathbf{z}_v) = \mathbb{E}_{\mathbf{x}^i \sim V}[\mathbb{1}[\max_{t \subseteq [1,T]} \mathbf{c}_k \cdot \mathbf{x}_t^i > e] = \mathbf{z}_v^i],$$

where $e$ is some constant. We then evaluate how well the set of discovered concepts $\mathbf{c}_{1:m}$ aligns with shapes 1 to 5:

$$\text{Alignment}(\mathbf{c}_{1:m}, \mathbf{z}_{1:5}) = \max_{P \in [1,m]^m} \frac{1}{5} \sum_{j=1}^5 \text{Match}(\mathbf{c}_{P[j]}, \mathbf{z}_j),$$

Figure 4: Nearest neighbors when applied to mixed_5c layer. The discovered concepts focus on smaller patches and captures lower level.

which measures the best average matching accuracy by assigning the best concept vector to differentiate each shape. For each concept vector $\mathbf{c}_j$, we test $\mathbf{c}_j$ and $-\mathbf{c}_j$ and choose the direction that leads to the highest alignment score.

**Creation of the Toy Example** The complete list of the target y is $y_1 = \sim (\mathbf{z_1} \cdot \mathbf{z_3}) + \mathbf{z_4}, y_2 = \mathbf{z_2} + \mathbf{z_3} + \mathbf{z_4}, y_3 = \mathbf{z_2} \cdot \mathbf{z_3} + \mathbf{z_4} \cdot \mathbf{z_5}, y_4 = \mathbf{z_2} \text{ XOR } \mathbf{z_3}, y_5 = \mathbf{z_2} + \mathbf{z_5}, y_6 = \sim (\mathbf{z_1} + \mathbf{z_4}) + \mathbf{z_5}, y_7 = (\mathbf{z_2} \cdot \mathbf{z_3}) \text{ XOR } \mathbf{z_5}, y_8 = \mathbf{z_1} \cdot \mathbf{z_5} + \mathbf{z_2}, y_9 = \mathbf{z_3}, y_{10} = (\mathbf{z_1} \cdot \mathbf{z_2}) \text{ XOR } \mathbf{z_4}, y_{11} = \sim (\mathbf{z_3} + \mathbf{z_5}), y_{12} = \mathbf{z_1} + \mathbf{z_4} + \mathbf{z_5}, y_{13} = \mathbf{z_2} \text{ XOR } \mathbf{z_3}, y_{14} = \sim (\mathbf{z_1} \cdot \mathbf{z_5} + \mathbf{z_4}), y_{15} = \mathbf{z_4} \text{ XOR } \mathbf{z_5}$.

We create the dataset in matplotlib, where the color of each shape is sampled independently from green, red, blue, black, orange, purple, yellow, and the location is sampled randomly with the constraint that different shapes do not coincide with each other.

**Hyper-parameter Choice and Sensitivity** To choose the hyperparameters, one can use a small-scale evaluation dataset to choose a few important hyperparameters. One should choose the hyperparamters so that they get concepts with high completeness and $R_1(\mathbf{c})$, and we better describe the impact of these hyperparamters to guide such selection.

**Choice of $\lambda_1, \lambda_2, \beta$:** We set $\lambda_1 = \lambda_2 = 0.1, \beta = 0.2$ for the toy dataset. We show the completeness score for varying $\lambda_1, \lambda_2, \beta$ in Figure 7,8,9 (when varying $\lambda_1$, we fix $\lambda_2 = 0.1$, and $\beta = 0.2$.) We see that both the completeness and alignment score are above 0.93 when $\lambda_1$ and $\lambda_2$ are in the range of $[0.05, 0.3]$, and $\beta$ is in the range of $[0, 0.3]$, and thus our method outperforms all baselines with a wide range of hyper-parameters. Therefore, our method is not sensitive to the hyper-parameter in the toy dataset. We set the $\lambda_1 = \lambda_2 = 0.1, \beta = 0.3$ for the NLP dataset, and we set $\lambda_1 = \lambda_2 = 10.0, \beta = 0$ for AwA dataset since the optimization becomes more difficult with a deeper neural network, and thus we increase the regularizer strength to ensure interpretability. The completeness is above 0.9 when $\lambda_1$ and $\lambda_2$ are set in the range of $[2, 20]$. Overall, our method is not too sensitive to the selection of hyper-parameter. The general principle for hyper-parameter tuning is to chose larger $\lambda_1$ and $\lambda_2$ that still gives a completeness value (usually $> 0.95$).

**Choice of $h, g$:** The intermediate layer (which effects $h$) can be chosen depending on the size of the nearest neighbor patch the user would like to visualize, since this size depends on the receptive field of the feature layer. Deeper layers have larger receptive fields while shallower layers have smaller receptive field size. In AwA experiments, we choose Mixed_5d layer so that the receptive field is $127 \times 127$, which is half of the original image size and capable of capturing both larger and smaller concepts. If the user is only interested in smaller and more low level concepts (such as the texture of image), we can apply our method to earlier layers. As a hyper-parameter control experiment, we apply our method to mixed_5c layer where the receptive field is $95 \times 95$ and visualize the result in Fig. 4, where indeed smaller (low level) concepts such as eyes (in constrast to head), furry, sandy are captured (which we name the concepts). If one finds the concepts discovered to be too small (low level), they could apply the method on a deeper layer and vice versa. For the choice of $g$, we let it be a two layer neural network (with 512 neurons) followed by the remaining network $h$, so that $h$ is also optimized in eq.3, which we fix in all experiments.

**Additional Nearest Neighbors for toy example** We show 10 nearest neighbors for each concept obtained by our methods and baseline methods in the toy example in Figure 10. The 10 nearest

neighbors for each concept obtained by different methods is used to perform the user study, to test if the nearest neighbors allow human to retrieve the correct ground truth concepts for each method.

**User Study Setting and Discussion**    For the user study, we set $m = 5$ (i.e. 5 discovered concepts) for all compared methods. The order of the 2 randomly chosen conditions (and which 2 conditions are paired), the order of the questions, the order of choices are all randomized to avoid biases and learning effects. All users are graduate students with some knowledge of machine learning. None of them have (self-reported) color-blindness. For each discovered concept, an user is asked to find the most common and coherent shape given the top 10 nearest neighbors. An example question is shown in Figure 15. Each user is given 10 questions, which correspond to the nearest neighbors of the discovered concepts for two random methods. (each method has 5 discovered concepts, and thus two methods have 10 discovered concepts in total). There are 20 users in total, and thus each method is tested on 8 users. For each method, we report the average number of correct answers chosen by the users. For example, if an user chooses shape 1,2,5,7,5, then the number of the correct answers chosen by the user will be 3 (since 1,2,5 are the ground truth shape obtained by the user). We average the correct answers chosen by 8 users for each method to obtain the "average number of correct answers chosen by users". We also report the average number of agreed answers chosen by the users. For example, if most users choose 1,2,5,7,5 for five questions respectively, we set 1,2,5,7,5 as the ground truth for the five questions. If user A answered 1,2,5,7,10 for the five questions respectively, his number of agreed answers would be 4. We average the agreed answers chosen by 8 users for each method to obtain the "average number of agreed answers chosen by users".

We find that other methods mainly fail due to (a) the same concept are chosen repeatedly (e.g. concept 2 and concept 4 of ACE). (b) lack of disentanglement (coherency) of concepts (e.g. concept 5 of PCA shows two shape in all 10 nearest neighbors). (c) highlighted concepts are not related to the ground truth concept (e.g. concept 4 of Kmeans). (a), (c) are related to the lack of completeness of the method, and (b) is related to the lack of coherency of the method.

**Implementation Details**    For calculating ConceptSHAP, we use the method in kernelSHAP [Lundberg and Lee, 2017] to calculate the Shapley values efficiently by regression. For ACE in toy example, we set the number of cluster to be 15, and choose the concepts based on TCAV score. For ACE in toy example, we set the number of clusters to be 150, and choose the concepts based on TCAV score. For PCA, we return the top $m$ principle components when the number of discovered concepts is $m$. For k-means, we set the cluster size to be $m$ when the number of discovered concepts is $m$, and return the cluster mean as the discovered concepts.

Figure 5: Nearest neighbors of top concepts by PCA.

Figure 6: Nearest neighbors of top concepts by Kmeans.

Figure 7: Completeness score and Alignment score for different hyper-parameter $\lambda_1$.

Figure 8: Completeness score and Alignment score for different hyper-parameter $\lambda_2$.

**Qualitative Results for Baselines in AwA**   We show nearest neighbors of the top concepts of PCA and Kmeans of the three class "Rabbit", "Squirrel", and "Weasel" in AwA in figure 5 and 6 respectively.

**Additional Nearest Neighbors for AwA**   We show additional nearest neighbors of the top concepts in AwA for all 50 classes from Figure 16 to Figure 24. For each class, the 3 concepts with the highest ConceptSHAP respect to the class with $R_1(\mathbf{c})$ above 0.8 is shown, along with the ConceptSHAP score with respect to the class. We see that many important concepts are shared between different classes, where most of them are semantically meaningful. To list some examples, concept 7 corresponds to the concept of grass, concept 33 shows a specific kind of wolf-like face (which has two different colors on the face), concept 27 corresponds to the sky/ocean view, concept 25 shows a side face that is shared among many animals, concept 46 shows a front face of cat-like animals, concept 21 shows sandy/ wilderness texture of the background, concept 38 shows gray back ground that looks like asphalt road, concept 43 shows similar ears of several animals, concept 31 shows furry/ rough texture with a plain background.

**Additional Nearest Neighbors for NLP**   We show additional nearest neighbors of the 4 concepts in NLP. The nearest neighbors of concept 1 and concept 2 are generally negative, and concept 3 and concept 4 are generally positive.

Figure 9: Completeness score and Alignment score for different hyper-parameter $\beta$.

Figure 10: (Larger Version) Nearest Neighbors for each concept obtained in the toy example.

Figure 11: (Larger Version) Nearest Neighbors for each concept for ACE obtained in the toy example.

Figure 12: (Larger Version) Nearest Neighbors for each concept for PCA obtained in the toy example.

Figure 13: (Larger Version) Nearest Neighbors for each concept for Kmeans obtained in the toy example.

Figure 14: (Larger Version) Nearest Neighbors for each concept for ACE-SP obtained in the toy example.

Table 3: The 4 discovered concepts with more nearest neighbors.

| Concept | Nearest Neighbors |
|---|---|
| 1 | poorly constructed what comes across as interesting is the<br>wasting my time with a comment but this movie<br>awful in my opinion there were \<UNK\> and the<br>forgettable \<UNK\> earn far more critical acclaim and win<br>wasting my time with a comment but this movie<br>worst 80's slashers alongside \<UNK\> with fear \<UNK\> deadly<br>worst ever sound effects ever used in a movie |
| 2 | normally it would earn at least 2 or 3<br>\<UNK\> \<UNK\> is just too dumb to be called<br>i feel like i was ripped off and hollywood<br>johnson seems to be the only real actor here<br>but this thing is watchable if only for bela<br>performance but they're all too unlikable to really care<br>way the fights are awfully bad done while sometimes |
| 3 | remember awaiting return of the jedi with almost \<UNK\><br>better than most sequels for tv movies i hate<br>male because marie has a crush on her attractive<br>think that about a lot of movies in this<br>i am beginning to see what she has been<br>cinema of today these films are the products of<br>long last think eastern promises there will be blood |
| 4 | new \<UNK\> \<UNK\> via \<UNK\> \<UNK\> with absolutely hilarious<br>homosexual and an italian clown \<UNK\> is an entertaining<br>stephen \<UNK\> on the vampire \<UNK\> as a masterpiece<br>and between the scenes the movie has one gem<br>make a film \<UNK\> in color light so perfectly<br>in it and the evil beast is an incredible<br>father and my adult son peter falk is excellent |

Figure 15: An example question of a screenshot of the human study.