[Reviews · NeurIPS 2020]

Review 1

Summary and Contributions: This paper proposes the concept of completeness which can be used to measure the effectiveness of concept-based explanation of deep neural networks. As the completeness measures the entire group of concepts, the authors additionally propose conceptSHAP which computes the marginal contribution of each concept. The authors used both quantitative and qualitative experiments to show the supremacy of their method compared to the classical ones including K-means and PCA. # updates after author response My questions are well addressed in the authors' response, I appreciate the authors for the clarification.

Strengths: I feel the rationale and theory development from the concept of completeness to conceptSHAP is well explained. I found completeness measure is universal to be applied in other tasks that are based on the clustering (see additional feedback section), thus I think the contribution is significant enough.

Weaknesses: Qualitative representation of AwA dataset was intriguing, but I feel k-means or PCA would yield similar concept gallery as they are shown to be able to retrieve relevant representations for certain classes in numerous previous researches, so I want to see the comparison on the qualitative side too in addition to quantitative completeness score.

Correctness: I found the claims and methods are correct.

Clarity: I found the paper is easy to follow.

Relation to Prior Work: This paper delivers detailed explanations of previous methods and clearly stated their direction of research: the measurement of concept-based explanation and its application, which is not found in previous contributions.

Reproducibility: Yes

Additional Feedback: Apart from the main focus of the paper: the explanation power of a deep neural network, I felt the method of concept-based explanation is very similar to that of clustering based representation learning as the activations are assigned to certain cluster centroid (or label, or prototype). Specifically, self-labeling and label transfer in [1] is a simultaneous version of concept discovering and assessing their contributions. Applying the concept of completeness to the task of self-labeling and other clustering-based representation learning would be interesting. [1] Asano, Yuki Markus, Christian Rupprecht, and Andrea Vedaldi. "Self-labelling via simultaneous clustering and representation learning." arXiv preprint arXiv:1911.05371 (2019).


Review 2

Summary and Contributions: The context of this work is concept-based interpretability which aims to explain algorithmic decision in terms of concepts underlying them. A crucial questions of this approach is how to discover or select these concepts. The paper proposes a measure for the completeness of a set of concepts which is inspired by the notion of sufficient statistics. This allows to define a loss function from which concepts can be obtained by optimization. Furthermore, the paper proposes an importance score for each concept which is based on SHAP. It analyzes the effectiveness of the proposed method in detailed experiments.

Strengths: I find the approach taken quite interesting. Sufficient statistics for concepts is quite natural and compelling. A notion of sufficiency allows for a comparatively well-motivated objective which can be optimized to obtain a set of concepts. Combining concepts with SHAP incorporates the theoretical motivations of Shapley values to concept-based explanations.

Weaknesses: The method depends on a number of hyperparameters such as 1.) the architecture of g in the completeness score (1) 2.) the number of concepts m 3.) \lambda_1 and \lambda_2 in the concepts objective (3) 4.) The layer at which the concepts are calculated 5.) The threshold \beta in L.111 The dependency on these hyperparameters is not (carefully) discussed. (The only exception is 3.) which is discussed in the Appendix.) For example, in L.101 it is stated that a layer for concept calculation is chosen "by starting from the layer closest to the prediction until we reached a layer that the user is happy with". Happiness strikes me as an extremely subjective and hardly quantitative measure. It would be great if the authors could elaborate on these hyperparameters in the rebutal with particular emphasis on 1.) and 4.). It is also not clear to me how the objective (3) for concept discovery is actually optimized. It appears to me that g will depend on the choice of c_1, ..., c_m. So for fixed c_i, one would need to find the optimal g and iterate this optimization after each update of the concepts. Could you please clarify this? Maybe, I have missed it but I could not find any discussion of computational costs of training with objective (3). This is a very useful and relevant piece of information. Could you please comment on this? I will adjust my score if these questions are adequately answered. ______________________________________________ ADDITIONAL COMMENT AFTER AUTHOR RESPONSE: The authors discussed the questions listed above convincingly. It would be great if the discussion could also be included in the manuscript and supplement. As a result, I have adjusted my score.

Correctness: The methods and claims are convincing and appear to be correct.

Clarity: The paper is well written and structured. I sometimes find the mathematical notation a bit sloppy but overall the presentation is very accessible and helpful.

Relation to Prior Work: The relation to previous work is carefully discussed. Similarities and differences are clearly stated. I should point out however that I do not have a full overview of the previous literature on concept-based explanations.

Reproducibility: Yes

Additional Feedback: L.94 "corresponds to the first 512 dimensions" -> "corresponds to the *last* 512 dimensions" L.121 As far as I can see, V is not defined. I assume it is the validation set. If so, probably one should not use the \sim notation but simply write x,y \in V as V is a set not a distribution. A similar comment applies to L.232. L.147: eta should be eta_f for consistent notation with definition (1). L.192: The approximation \int p(y|z)p(z|x) \approx p(y | E(z|y)) seems extremely poor unless the variance of the distribution p(z|x) is small. Is there any reason to believe that this is the case? L.199: Missing space after "different as possible." L.208: In equation: x^b_a \subset T_{c_k} should be x_b_a \in T_{c_k}. l.232: It would be helpful to write \eta_j(c) = ... in equation 4. l.234: \hat{g} = argmax g is confusing notation. If you use mathematical notation, please specify what argument the argmax is acting on. Also g should depend on the argument of the argmax. Probably, it is better to simply state in words that this is the map g obtained by optimization.


Review 3

Summary and Contributions: This paper proposes doing automated concept discovery for interpretability by optimizing to find vectors in a CNNs activation space which are sufficient to estimate the model's prediction while being interpretable. The main contribution over previous work is to formalize both human interpretability and model explanation fidelity (termed 'completeness') of concepts in such a way that it is optimizable directly.

Strengths: 1. The paper formalizes the objective of previous work (ACE) which allows for the direct optimization of this objective. This formalization is important not just for the empirical improvement, but also by clarifying why we should trust the explanations offered by the method. The method is trustworthy, because it is directly optimizing to maximize explanation fidelity. The method is human-usable because it directly optimizes for enabling ostensive definition via the top-K loss. 2. The paper shows considerable improvement over previous work across both benchmarks considered both in terms of fidelity (aka completeness) and human interpretability. This strength is however limited because it remains unclear to me why the authors validated on these datasets instead of imagnet as was done for previous work on CAVs. 3. The paper extends shapley values to apply to the concept discovery setting, although minor this contribution is important for showing how previous ideas on interpretability extend in a straight-forward way.

Weaknesses: 1. (Partially addressed by rebuttal) As mentioned above, I am somewhat concerned about the generality of this method. The method was tested on non-standard datasets for unclear reasons, and there is no layer choice hyper-parameter ablation study. 2. (Addressed by rebuttal) The human interpretability experiment set up seems interesting, but a bit strange to me. Why not show the test subject a subset of the method's most concept/cluster relevant images and then ask them to classify some other images as concept/cluster relevant or not relevant (the set-up I'm suggesting is used by contemporary work submitted to NeurIPS 2020)? Importantly, this should be done on the AwA dataset or Imagenet, not just the toy dataset. 3. (Addressed by rebuttal) The NLP experiment augmenting inputs is an interesting idea, but it seems to me it needs a control scenario. For example, why not append 5 random words to check whether the effects of going out of distribution is somehow having a large effect? Overall, I think the motivation and proposed solution are great, but the empirical validation is somewhat problematic. A priori, if I had only read the description of the method without the experiments, I'd suspect that the method would be unlikely to provide human interpretable concepts. Although the experimental design could feasibly deal with this concern, the incomplete benchmarking raises questions for me about how the experiments were selected and why results were not more extensively reported.

Correctness: I have a few specific, mostly minor, questions and criticisms: (Addressed by rebuttal) 117-119: Unclear to me why this is using x_t and not just x. For a convnet without adaptive pooling it seems to me that f(x_t) is not even defined. Perhaps this also needs to be changed in l.129 Also, why should we treat x_t independently? Would it perhaps make sense to do a global average or max pool and then apply the proposed method to the pooled vector? (Addressed by rebuttal) 140: This example seems like more of a pathological example than proto-typical. Usually a subset of concepts or activations provide partial information, and you want a score that reflects this fact. In fact the proposed score does that, but this example doesn't demonstrate it. More importantly this example raises the question to me, in situations like the XOR, it seems like the concepts are in some sense a narrow optimum and this might pose a challenge to optimizing for concept discovery. If this concern actualizes, will the method generalize to other datasets, models and layers?

Clarity: The paper is very clear. The point I mentioned above about x_t vs x should be clarified. Another minor comment, it seems to me that the top-K loss encourages human interpretability by allowing ostensive definition. Is this the authors' intention? If so, it would be nice to state that in the paper, and possibly have a footnote or appendix commentary on why this loss corresponds to some human interpretability desiderata.

Relation to Prior Work: Yes, the relation to prior work is clearly explained.

Reproducibility: Yes

Additional Feedback:

[Author Response · NeurIPS 2020]

We thank all reviewers for their constructive comments. We first address some general questions, and then respond to the questions raised by each reviewer separately.

**Layer Choice (R2,R3)** The layer can be chosen depending on the size of the nearest neighbor patch the user would like to visualize, since this size depends on the receptive field of the feature layer. Deeper layers have larger receptive fields while shallower layers have smaller receptive field size. In AwA experiments, we choose Mixed_5d layer so that the receptive field is $127 \times 127$, which is half of the original image size and capable of capturing both larger and smaller concepts. If the user is only interested in smaller and more low level concepts (such as the texture of image), we can apply our method to earlier layers. As a hyper-parameter control experiment, we apply our method to mixed_5c layer where the receptive field is $95 \times 95$ and visualize the result below in Fig. 1 top right, where indeed smaller (low level) concepts such as eyes (in contrast to head), furry, sandy are captured (which we name the concepts). If one finds the concepts discovered to be too small (low level), they could apply the method on a deeper layer and vice versa.

**Qualitative Results for Baselines and User study on AwA (R1,R3):** We have added qualitative examples (see Fig. 1 bottom row) for concepts discovered by PCA and Kmeans on AwA (ACE gives the same set of concepts as Kmeans if superpixels are replaced by squares), where we choose top concepts by conceptSHAP (in the same way as our method exactly) for the classes "squirrel, rabbit, bobcat". To evaluate our method, PCA, Kmeans (ACE is Kmeans without superpixel), we added a human study suggested by R3 on AwA. For each method, we randomly choose 1 top concept per class for the 3 classes, and thus we choose 3 concepts per method (9 concepts in total). For each concept, we show users 4 top images of that concept, and ask users to choose the image (out of 3 different options) that they believe should belong to the same concept (where one of the option will actually belong to the same concept, and the other two are random image patches of the same class that does not belong to that concept). We then calculate the average accuracy to measure the human interpretability of the concept discover method. We conduct a user study with 10 users, where each of them are asked with the same 9 questions (1 question per concept chosen). The average correct ratio for our method, PCA, and Kmeans are 0.733, 0.267, and 0.6 respectively, showing our method's superiority. Kmeans outperforms PCA as it also encourages closeness of top nearest neighbors (which is better for ostensive definition).

**Hyperparameters (R2):** To choose the hyperparameters, one can use a small-scale evaluation dataset to choose a few important hyperparameters. One should choose the hyperparamters so that they get concepts with high completeness and $R_1(\mathbf{c})$, and we better describe the impact of these hyperparamters to guide such selection. We provide experiments on sensitivity to hyperparameters in the appendix: to the number of concepts in Fig. 2, and to $\lambda, \beta$ in Figs. 4, 5 and 6 in Appendix. We fixed the architecture $g$ in our previous experiments to a two layer neural network. It can also be a linear network followed by the remaining neural network $h$ (so that $h$ is also optimized in eq.3 in line 209). We show the qualitative results of AwA where $g$ includes $h$ (which we call "ours-linear g+h") in Fig. 1 (top left), where we obtain interpretable concepts (we named the concepts) with high completeness (thus the hyperparameter is not sensitive to architecture if $g$ has enough representative power). We discussed how to select the layer choice and its impact above.

**Optimization, Computation (R2):** To optimize the objective $\arg\max_{\mathbf{c}_{1:m},g} \log P(h_y(g(v_\mathbf{c}(x_{1:T}))) + R(\mathbf{c})$, we optimize variables $\mathbf{c}_{1:m}, g$ jointly. When the optimization converges, $g$ is a (local) optimal value given $\mathbf{c}_{1:m}$. The computational cost for discovering concepts and calculating conceptSHAP is about 3 hours for AwA dataset and less than 20 minutes for the toy dataset and IMDB, using a single 1080 Ti GPU, which can be further accelerated with parallelism. The computational cost is low since the pretrained model is fixed, and we only optimize for $g$ and $\mathbf{c}$.

**R3:** The point in lines 117-119 pointed out by the reviewer is indeed a typo, it should be $x$ instead of $x_t$, we have corrected it. While XOR may make the ground truth concept difficult to discover, in our toy example we show that our method can still retrieve the correct concepts with XOR in the model. Since we show that our method can work on both toy and real datasets, we believe this further demonstrates the generality of our method even when the optimization may seem difficult. The top-K loss does encourages human interpretability by allowing ostensive definition, which we will better discuss. For the NLP experiment, we added a control scenario where we append 5 random subsentences, and the average prediction score becomes 0.498 (oiginally 0.516). Thus, appending discovered concepts experiments is not caused by being out-of-distribution. We didn't test on Imagenet since we can't visualize results for all 1000 classes.

Figure 1: Visualizing ours-linear g+h (top left), ours-mixed_5c (top right), PCA (bottom left), Kmeans (bottom right).

[Meta-Review · NeurIPS 2020]

I recommend accepting this paper. I believe that this work will be of relevance to the machine learning community. Although reviewers initially had some doubts about a few aspects of the methodology, they were convincingly addressed in a rebuttal.